# Sounding Mechanism of a Flue Organ Pipe—A Multi-Sensor Measurement Approach

**DOI:** 10.3390/s24061962

**Published:** 2024-03-19

**Authors:** Paolo Bordoni, Piotr Odya, Józef Kotus, Bożena Kostek

**Affiliations:** 1School of Industrial and Information Engineering, Polytechnic University of Milan, 20156 Milan, Italy; paolo1.bordoni@mail.polimi.it; 2Multimedia Systems Department, Faculty of Electronics, Telecommunications and Informatics, Gdańsk University of Technology, Narutowicza 11/12, 80-233 Gdańsk, Poland; pioodya@pg.edu.pl (P.O.); joseph@multimed.org (J.K.); 3Audio Acoustics Laboratory, Faculty of Electronics, Telecommunications and Informatics, Gdańsk University of Technology, Narutowicza 11/12, 80-233 Gdańsk, Poland

**Keywords:** pipe organ, multi-sensor measurement, numerical simulation

## Abstract

This work presents an approach that integrates the results of measuring, analyzing, and modeling air flow phenomena driven by pressurized air in a flue organ pipe. The investigation concerns a Bourdon organ pipe. Measurements are performed in an anechoic chamber using the Cartesian robot equipped with a 3D acoustic vector sensor (AVS) that acquires both acoustic pressure and air particle velocity. Also, a high-speed camera is employed to observe the jet coming out from the windway. For that purpose, the steam resulting from dry ice and hot water is used. A numerical simulation of the sounding mechanism of a pipe of the same geometry is based on measuring the pressure signal and the intensity field around the mouth employing an intensity probe and visualizing and observing the motion of the air jet, which represents the excitation mechanism of the system. The ParaVIEW software serves for the simulation and visualization of the air jet. Then, the results obtained from measurements and simulations are compared and discussed. Also, some future directions discussing the application of a machine-learning approach to the area of pipe organ air flow investigation are contained in the Conclusions section.

## 1. Introduction

An investigation into the acoustics of organ pipes has become a field of intensive research based on several scientific areas, such as aerodynamics and vibroacoustics, over several decades [1,2,3,4,5,6]. The examination of phenomena occurring in flue organ pipe sound production also provides information for wind instruments in general due to their less critical dependence on the musician’s interaction and the basically constant air flow that constitutes the input of the pipe system. Along with theoretical investigations, new opportunities appear based on state-of-the-art programs to observe and investigate the mechanisms influencing pipe sound characteristics. They also provide insight into the validation and refinement of analytical models. More specifically, numerical simulations have become a prominent framework for a closer investigation into the nonlinear interactions between the air jet motion and the air column in the resonator, providing a way to investigate the complex energy exchange between the fluid dynamics of the jet and the acoustic field generated [3,7]. The main advantage lies in the possibility of comprehensively analyzing the physical quantities involved without dealing with the intrinsic problems related to measurements in situ or anechoic chamber conditions, especially in micro-scale observations. Nevertheless, such simulations are limited by necessary approximations that prevent them from fully overlapping with measurement results.

In the current literature, besides the studies performed directly in the numerical world [8,9,10,11], fewer publications explore matching between the numerical simulation and the behavior of actual pipes [4,12,13], mainly from the resulting pressure signal perspective. Hence, a comprehensive qualitative validation of the numerical simulation technique applied to flue pipes is an area worth exploring.

The present work aims to examine how to evaluate the reliability of a numerical simulation of an organ pipe by comparing it with the results of the measurements conducted on an actual pipe of the same geometry. Due to limitations in observing sound production phenomena, measurements were performed on the external part of the sounding mechanisms. In this way, the simulation results could be adapted and processed to be directly compared with the measurement outcome. More precisely, three perspectives were pursued. One focused on the sound radiating from the pipe, reconstructing the intensity fields around the mouth region and employing the Cartesian robot and a 3D acoustic vector sensor (AVS) in an anechoic chamber. The second, related to the former, was based on the harmonic content of the pressure signal, measured in the near field. The last was directed towards the visualization of the motion of the air jet, which involves a high-speed camera and a special setup for making the air jet visible. The limitation of experiments performed was also examined, and the outcomes turned out to be suitable not only for comparison with the numerical simulation results but also for inquiries between the different perspectives of measurement. Figure 1 shows an outline of the study goals.

The paper is structured as follows. The following section is devoted to a brief description of the instrumentation for retrieving data and extracting information from measurements on actual pipes, and the fundamentals of numerical simulations are also described. 

Section 3 focuses on the description of the implementation of the model, the running of the simulation, and the results. The measurements presented in Section 4 are based on an actual organ pipe of the same geometry. The measurement setup is illustrated, along with the post-processing steps applied to the collected information.

In Section 5, the results obtained by the simulation, as well as the two different types of measurements, are compared, highlighting the correspondence but also the differences, which might be addressed by further research. Finally, Section 6 summarizes the findings together with some remarks on the limits of these approaches, both in the simulation and in the measurement stages. Also, future research directions applying machine learning for the pipe organ investigation are outlined. 

## 2. Sound Production in Organ Pipes

Organ pipes are conventionally split into the following two categories: labial (flue) pipes and lingual (reed) pipes. Both can be present in the same instrument, but they belong to a different set of ranks. Since only labial pipes are in the modeling and measurements of this paper, this typology is described in all its parts (Figure 2). From the bottom to the top, air flow first meets the pipe foot, that is, the conical section that goes from the foot hole to the column-shaped upper part; this one is referred to as a resonator or pipe body. At this level, a horizontal plate (the languid) forces the air to move through a thin gap (flue or windway) in correspondence with the lower lip of the mouth. The mouth is delimited on both sides by the ears (if present) and on the top by the upper lip. The rest of the pipe is made of a resonator that can have a constant cross-section or different shapes. Finally, on the top of the pipe body, one can find the stopper (for stopped pipes) or a tuning slot (for open pipes) devoted to adjusting the effective (acoustic) length of the resonator. 

Therefore, according to the abovementioned subdivision of pipes (open and stopped), the most important properties of the pipe sound can be presented. First of all, looking at the sound spectrum of any organ pipe, one can identify a series of peaks as follows: the lowest is referred to as the fundamental frequency, while the others are called partials, and those which are in harmonic relation with the fundamentals take the name of overtones (Figure 3). The shape obtained linking the peak values of the partials represents the spectral envelope, and it is strongly related to the perceived timber of the sound [14,15]. Even though pipes are considered sustaining instruments (that is, a note can be held with a constant loudness), the Attack–Decay–Sustain–Release envelope model still applies, and it is meaningful to investigate, especially in the attack phase, since the relationship between partials can significantly change before they settle [1] and, in some cases, it can be controlled when played on [16].

**Figure 2 sensors-24-01962-f002:**
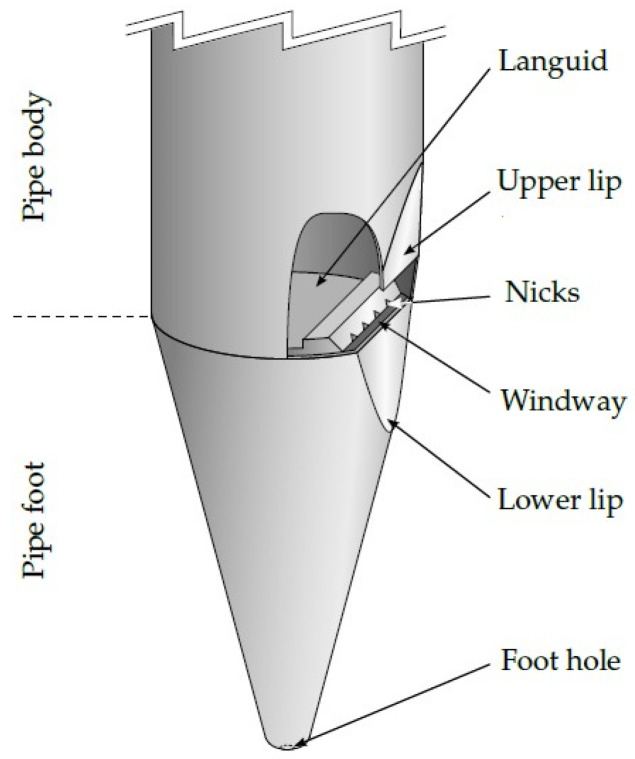
A scheme representing the parts of a flue organ pipe [17].

Investigations into the sounding mechanisms of organ pipes became a topic of scientific publications starting from the 20th century, although handbooks and manuals devoted to the building of an organ date back to the 18th century.

A good summary of the literature up to the year 2000 is found in the paper written by Miklós and Angster [1]. This work gives an overview of the process of sound generation in flue pipes as a whole, but at the same time, it underlies the different roles of the three subsystems involved:The air jet blowing from the flue and hitting the upper labium (excitation mechanism—aerodynamic system);The air column inside the pipe is excited by the jet (acoustic resonator–acoustic system);The pipe wall is set into motion by the acoustic waves (vibration–mechanical system).

**Figure 3 sensors-24-01962-f003:**
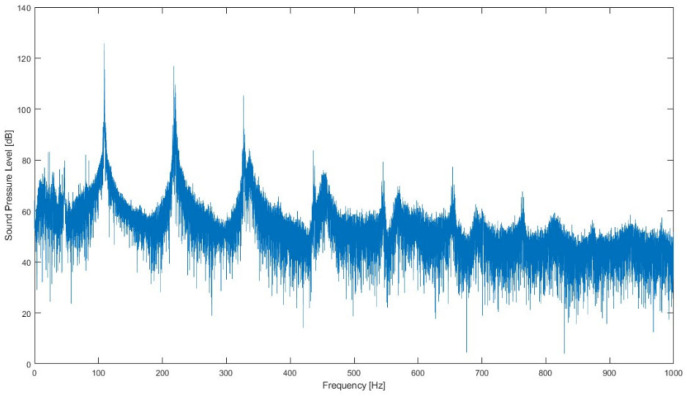
Sound spectrum of an open organ pipe (obtained from the measurement performed).

Moreover, it can also be claimed that, by the end of the 20th century, research was mainly developed in the following two areas: the theoretical approach based on physical theory and formulas [18,19,20,21,22] and a practical approach focusing more on measurements and the analyses of experimental data [23,24,25,26,27,28,29,30]. In recent years, thanks to a massive growth of computational power and the development of new software, the third thread of research became prominent, namely the numerical simulation and modeling of aerodynamics as well as acoustic and mechanical systems [29,31,32]. Based on the spatial and temporal discretization of the governing equations, these virtual models enable the retrieval of some information about physical quantities that cannot be measured in a real context, for example, because the presence of a probe could alter the real working conditions and, therefore, it would give a misleading value. Moreover, they make it possible to evaluate, to a larger extent, the mathematical models known today and make comparisons to experimental records. Lastly, parametric simulations are crucial for applying optimization algorithms to nonlinear problems to which the pipe subsystem’s interactions belong [17,32].

The application of numerical simulations to research on the fluid dynamics and sound radiation of organ pipes has gained great popularity in recent years. The most interesting area is the edge tone formation at the upper labium, which is valid not only for organ pipes but also for many other wind instruments. Vaik and Paál performed detailed parametric investigations concerning the velocity, the inlet profile, and the jet-edge distance [7]. More specific adjustments, namely the height of the lower labium relative to the languid and the inner geometry of the foot hole, were investigated by Adachi et al. [9]. A deep study on the whole foot model is also present in the literature, with important observations on the jet’s natural oscillation, its velocity profile, and some optimum conditions for a stronger and more stable edge tone [33,34]. Fischer et al. proposed the simulation of the entire pipe as well as some surrounding areas, reproducing not only the motion of the jet but also the sound wave propagation from the mouth and its pressure level spectra [35]. A very similar approach was exploited for earlier theories on the aerodynamic sound source [10,11,36].

Nowadays, considerable effort has been put into confirming or dispelling some rules-of-thumb employed by organ builders, and the overall knowledge of the effect of many parameters has been improved. Other investigations have tried to provide a better understanding of the interaction between the jet, acoustic resonance, and sound radiation.

For both observations as well as numerical simulations on the air jet, a two-dimensionality assumption has always been taken. This can match the reality in many cases, but, especially the jet, it is sensible to many variables and can be influenced by asymmetries or irregularities along the width of the flue, as underlined by Rucz et al. [31]. For example, it has been observed that adding some nicks to the languid makes the air jet more stable [3]. The 2D assumption must, therefore, be validated (or possibly disproved) through observations of the air jet from a different perspective and a fully three-dimensional numerical simulation. In general, a 2D model reduces the computational cost, but at the same time, it can offer a method of comparing with a real pipe only by means of the generated pressure fluctuation. For a multi-perspective analysis of the numerical simulation outcome, a full 3D model is needed, and such a model was created for the experiments described in this manuscript.

## 3. Numerical Model

The proposed model has been realized using some modules of the OpenFOAM-v2012 C++ toolbox and free and open-source software released under the GNU general public license [37]. The software supports the Large Eddy Simulation (LES) mathematical model, which is one of the most promising and successful methodologies for simulating turbulent flows [2,38]. It is a very useful tool in aeroacoustics because, in many problems, noise and sound are generated mainly through large-scale fluctuations, which are computed directly into the LES. If properly implemented and validated, LES codes can simulate flow physics accurately enough to capture energy transfer from turbulent to acoustic modes. It should be noted that in the context of fluid dynamics and turbulence, “large-scale fluctuations” refer to variations or changes in fluid properties (such as velocity, pressure, and temperature) that occur over relatively large spatial distances within the flow field. These fluctuations are typically associated with the larger eddies or structures present in the turbulent flow.

The model has been designed according to the geometry of the stopped Bourdon organ pipe available at the Multimedia Systems Department of the Gdansk University of Technology (GUT), Gdansk, Poland. This stopped pipe is depicted in Figure 4. It has an overall length of 820 mm, of which 80 mm is the height of the foot, and the remaining 740 mm comprises the resonator. It is 75 mm wide and 88 mm deep. The walls are 10 mm thick in all directions; therefore, the column air has a cross-section of 55 × 68 mm. The mouth has a height of 18 mm, and the labium becomes thinner approaching it until it reaches 1 mm at the tip. The windway has the same width as the inner pipe, and it is 0.5 mm deep.

With the help of the software, the mesh was generated inside a 3D volume resembling the internal air column of the resonator, together with a portion of the free space just right outside of the mouth. The model aims to simulate the behavior of a stopped organ pipe; therefore, the presence of the stopper is modeled as well, so the air volume results are 710 mm long.

The mesh grid was designed to make the jet simulation more accurate while leaving the upper part of the resonator and the external volume coarse. Using the same refinement on the whole domain dramatically increases the computational time needed to run the simulation. Hence, at the windway, the cell length is 0.25 mm, whereas at the top of the resonator and in the outside mouth region, cells can reach the size of about 5 mm. Details concerning the mesh grid are explained further on.

The 3D model was carried out by extruding the geometry in the z-direction and exploiting the symmetry of the geometry. Hence, the profile described above turned out to be the pipe’s sagittal section, while the left side was superimposed on it at a distance of 27.5 mm in the z-direction, that is, the half-width of the pipe. The overall mesh is assembled by exploiting the blockMesh utility available within the OpenFOAM package. Since the geometry of the pipe can be reduced as a combination of cuboids, it can be implemented using the following hexahedron objects (hex):The main resonator (block A), proceeding from the tip of the labium up to the stopper;The mouth region (block B) with the remaining part of the resonator at the mouth level;The jet region (block C), linking the windway to the tip of the labium;The external mouth region (block D) and an external labium region (block E) are separated from the rest of the subsequent blocks because they are still in contact with the side walls of the pipe;The free space blocks F, G, H, I, J.

Blocks F and H touch the external surface of the pipe body, namely the so-called beard (front face of the pipe foot). F, G, H, I, and J are extended until z = 50 for modeling the free air space outside the pipe in the neighborhood of the mouth.

For each block, it is possible to define the number of cells in which the corresponding hexahedron is divided along the three dimensions. Moreover, one can set the size of the cells to be either constant or stretched/shrunk in one direction. Taking advantage of these functionalities, the mesh grid is designed to make the jet simulation more accurate while leaving the upper part of the resonator and the external volume coarse. Using the same refinement on the whole domain dramatically increases the computational time needed to run the simulation. Hence, in block C, the smallest cell length result is 0.25 mm, whereas, for blocks B, C, and D, a cell size of about 1 mm is set. Finally, in blocks A, E, F, G, H, I, and J, cells can reach a size of about 5 mm while moving away from the jet region. The resulting mesh model comprises 201,534 points, 176,600 cells, and 554,229 faces, of which 505,371 are internal and 48,858 are external.

For the definition of boundary conditions, the encountered physical boundaries are collected into three groups, namely the solid walls of the pipe (the wall boundary condition), the inlet at the windway (the inlet), and the free space outside the mouth (the outlet). In addition, the symmetry of the simulated domain is exploited, halving the overall computational cost by means of the symmetry boundary condition. The settings are listed in Table 1, together with the initial conditions of the internal field. The simulation time step was set to *δt* = 2 × 10^−5^ s, ending at *t* = 0.5 s.

The simulation was run on a server equipped with a Threadripper 2950-X CPU available at Gdansk University of Technology. The process lasted about 23 h and produced approximately 400 MB of data. The fields of interest were the velocity *U* (i.e., volume acoustic flow) and the pressure *p*, and they were stored every 50 ms of the simulated time (i.e., every 2500 simulation steps). Furthermore, six virtual probes were set along a horizontal line 2 cm outside the pipe spanning the mouth width. The resulting datasets were visualized and inspected with the ParaVIEW v5.9 software according to a customizable perspective, level of detail, and color scale (see Figure 5).

## 4. Measurements

The organ pipe was analyzed from two main perspectives, namely audio and video. In relation to the audio-based analysis, a 3D acoustic vector sensor (AVS; USP—Ultimate Sound Probe) was employed. Its location in space relative to the pipe was accurately set thanks to a Cartesian robot. It is a *p* − *u* type measurement, i.e., the pressure *p* (scalar) and particle velocity *u* (vector) are measured simultaneously. The data acquired enabled the reconstruction of the sound intensity vector at any point in space. The probe was appropriately calibrated [39].

The robot is located inside the anechoic chamber of the Multimedia Systems Department of Gdansk University of Technology (GUT). The role of the Cartesian robot is to position the acoustic probe precisely [39]. Its design guarantees the fixed space orientation of the acoustic probe. The robot can place the measurement probe with a positioning accuracy of 200 [μm]. Its ranges of motion are X—1850, Y—2000, and Z—1540 [mm]. The movement speed for a particular axis is as follows: X—50, Y—50, Z—15 [mm/s]. Due to the design of the robot’s arm, the horizontal placement of the pipe has been adopted: the probe, attached to the arm from above, is then capable of moving even within 1 mm of the pipe (Figure 6). To avoid unwanted interference in the recordings, the compressor providing the constant air pressure for the long-term steady-state response of the pipe was placed outside the anechoic room and connected to the pipe stand with a 15 m long hose.

The robot’s arm probes a volume of 10 × 15 × 3 cm outside the pipe’s mouth, with a step of 1 cm, collecting data in 638 points. The recordings are 1.5 s long and stored as binary files. Each of them contains five channels at the sample rate of 48 kHz: the first is a synchronization channel (not used), the second one is the pressure signal, and the remaining three are the particle velocity signals in the three orthogonal directions.

The intensity field was computed in the MATLAB R2022b environment and converted into ParaVIEW software for 3D visualization. Figure 7 shows the results from different perspectives.

The visualization of the air flow was obtained with a high-speed camera (HSC) and required a special setup to make the jet coming out from the windway visible. It was chosen to exploit the steam produced by the reaction of dry ice (carbon dioxide in its solid form) and hot water. This choice was adopted thanks to its simplicity and ensuring safety conditions in the measurement environment. The pressure produced by the expansion of dry ice pills during the sublimation into the water was sufficient to drive the organ pipe to its first resonance mode for a few seconds. Therefore, the compressor was not used.

HSC and lights were placed according to two different arrangements. The first one encompassed positioning HSC directly in front of the pipe’s mouth. The two lights were installed on the left and right sides, respectively (Figure 8), in such a way that the transverse orientation of the light beams enhanced the visibility of the jet. The second arrangement focused on the transversal motion of the jet, with the camera placed at an angle of about 80° on the right side of the pipe and two lights in front of the pipe.

The camera was set to take clips of about 4 s at 2000 frames per second (fps), which, when reproduced with a standard digital video frame rate (30 fps), resulted in a speed factor of 0.015.

**Figure 8 sensors-24-01962-f008:**
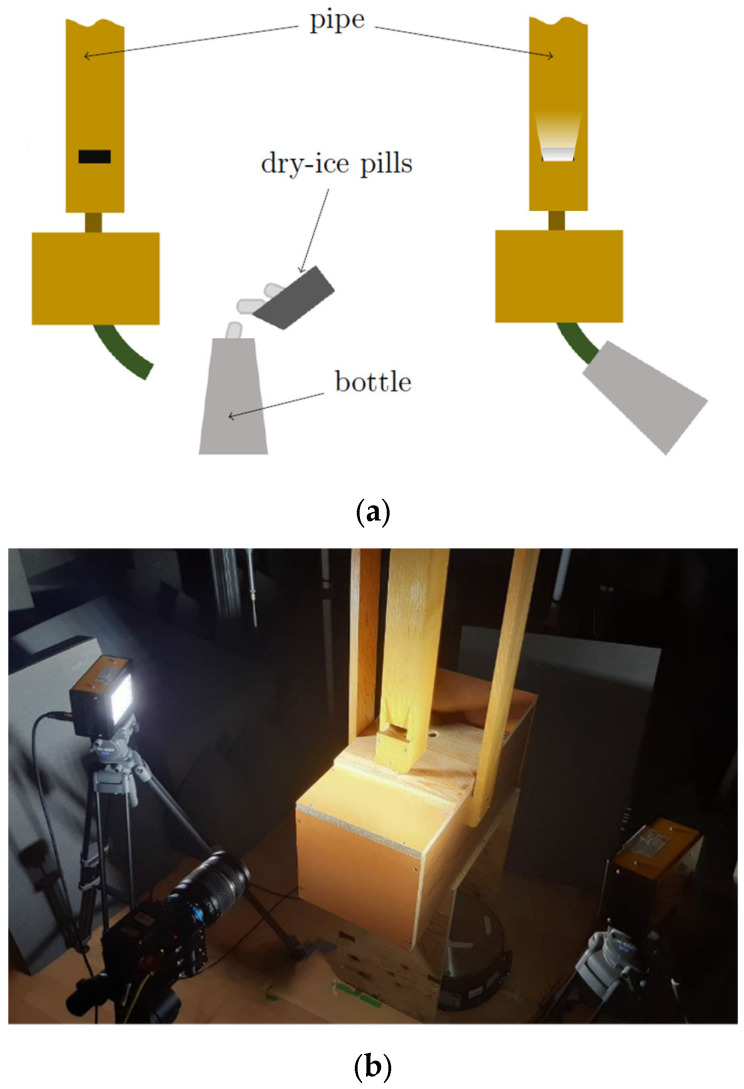
High-speed camera setup: (**a**) measurement schema, (**b**) measurement visualization with a high-speed camera (HSC). The pipe, lights, and camera are placed to enhance the visibility of the air jet motion.

## 5. Results and Discussion

The collected data on the real organ pipe enabled us to make a qualitative evaluation of the validity of the numerical simulation. With respect to the pressure signal, a comparison between the sound spectrum detected in the vicinity of the mouth and the spectrum of the simulated pressure signal at the very same place is shown in Figure 9. When compared with the simulated signal, it can be observed that the simulated fundamental frequency (102 Hz) shows good agreement with the real one (104.9 Hz). Moreover, the simulation models the sequence of the pipe’s resonance modes rather than the sound’s harmonics. Such a relation holds up to the VIIth mode, and the smallest errors occur at the Vth mode. Table 2 illustrates the error that occurs between the measurement assessments simulation. The differences between mode frequencies measured and obtained in a numerical simulation suggest that the model reproduces the key acoustic characteristics rather well.

Unfortunately, some artifacts are visible in the analysis (see Figure 9). It may be related to the noise-like component of the measured spectrum being dampened in the simulations.

Moreover, some discrepancies can be noticed. First, the harmonic series of the aeroacoustics sound source of the labium is not present. The model introduces damping in the vortex-shedding around the labium tip, preventing the simulation from capturing the acoustic source associated with the unsteady vortex dynamics. Conversely, some unexpected frequency peaks reveal some undesired resonances inside the model, which might be explained by the non-perfect wave-absorbing condition modeled at the free-space boundaries of the outside-mouth volume as they do not correspond to any acoustic mode visible in the measured signal. Moreover, due to time constraints, the mesh was left too coarse, so this might have prevented the reconstruction of small edges. 

**Table 2 sensors-24-01962-t002:** Comparison between the measured and simulated frequency value of the first seven acoustic modes of the resonator.

Acoustic Mode	Measurement	Simulation	Error
I	104.9 Hz	102 Hz	2.76%
II	314.8 Hz	318 Hz	1.02%
III	536 Hz	540 Hz	0.75%
IV	769.4 Hz	766 Hz	0.44%
V	999.1 Hz	994 Hz	0.51%
VI	1241 Hz	1226 Hz	1.21%
VII	1475 Hz	1456 Hz	1.29%

Another way to compare the simulation outcome and the measurements comprises the sound intensity distribution around the mouth of the pipe. For better visualization, only the intensity values along the z-axis are considered and with a step of 5 mm. The intensity distribution is displayed in Figure 10. The contours reveal the interesting distribution of the sound intensity, especially when very close to the pipe labium. A greater intensity at the sides of the mouth is observed, while in the center, it shows a value of approx. 30% lower. The plot seems to suggest that the sound production is more intense in the place where the labium joins the lateral pipe walls. In the distribution provided by the simulation, such a phenomenon is not observed. It might be that the windway is a bit narrower at the center, resulting in a weaker jet in that position. These details were not caught by the simulation since the symmetry of the pipe was one of the initial hypotheses during the building of the model and the running of the numerical simulation.

The third way to analyze the simulation outcome is the comparison with the visualization of the air jet. For an easier juxtaposition, the velocity field representation in the ParaVIEW software is arranged together with the 3D representation of the pipe walls. The field is set according to a scale of white shades, with opacity proportional to the velocity. This way, a representation close to the one captured by the HSC is obtained (Figure 11). Focusing on the lower part of the jet, where the air sheet motion can clearly be observed in the HSC recording, we can notice how the simulation shows a different shape of deflection. More specifically, it seems that the jet moves straight upwards until about half of its way and assumes a transverse displacement only downstream of that point. On the contrary, the clip frames show that the jet deflects right above the flue exit, according to a fully curved path shape. The reasons behind this discrepancy might lie in the faulty modeling of the interaction between the air jet and the acoustic field crossing the mouth region or in an erroneous jet velocity value set in the numerical simulation.

Furthermore, in the video recordings, some strands reveal the not-so-perfect flat surface of the windway walls, which makes the jet thicker at some points and thinner at others. Due to the perfectly flat surfaces provided by the model, none of these irregularities appear in the numerical simulation (see YouTube link: https://youtu.be/WlrlTKOpdFo, accessed on 9 March 2024).

The model can also be exploited for a wide range of visualization perspectives. Even in a coarse mesh simulation, like the one proposed in this work, while looking at the jet’s profile from the labium perspective (Figure 12), the sheet shape turns into a wavy motion in the z-direction. The comparison between the jet visualization and the intensity field distribution offers some other clues on the three-dimensionality of the jet. The frontal video recording shows that the steam motions appear to be more pronounced at the sides where the labium is joined to the pipe walls than compared to the middle. The larger motion of the air sheet is in agreement with the sound intensity field distribution (Figure 12), which indeed shows a larger value close to the pipe walls. However, it must be further explored whether this is detectable in any other wooden pipes or is a specific feature of the pipe used in the current experiments.

## 6. Conclusions

The numerical LES model of the Bourdon organ pipe has been compared from different main perspectives. One focused on the sound generated by the pipe, with the reconstruction of the intensity field around the mouth and the acquisition of the pressure signal. The second visualized and recorded the motion of the air jet, with a detailed description of the effort put into making it visible. It was possible to correctly model the resonance modes of the pipes in the simulation, but not the harmonic sound generated at the labium tips. This issue may be overcome with the refinement of the mesh, especially in the mouth region, which, however, translates into an increase in the computational cost.

The bell-shaped profile of the intensity sound is in good agreement with the measured one in the near field, but it does not capture the specific feature of an increased sound intensity emission at the sides of the mouth. Measurements on other pipes may clarify if this property belongs only to the pipe used in these experiments or is valid for all wooden pipes.

The visualization of the air jet shows a non-perfect overlap with the simulation result due to the absence of some imperfections on the flue walls that have a large influence on the real jet behavior. However, even for geometry with perfectly flat surfaces, such as the one from the model used, a jet horizontal curvature can be observed, revealing the non-2D motion of the jet.

The limitation of the results, concerning both the numerical simulation and the real pipe measurements, leaves space for some questions/hypotheses that should be addressed in further research. 

Also, this is one of the few areas that machine learning did not pass through. However, several future directions of applying machine learning to investigate pipe organ flow air may be envisioned. This may concern several regions. One of the possible applications is developing machine learning algorithms to automatically detect and characterize imperfections in flue walls that influence air jet behavior. This could involve training models on data obtained from simulations and measurements to identify patterns indicative of imperfections. In contrast, exploring machine learning techniques to improve the accuracy of numerical simulations by incorporating data-driven corrections or adjustments based on real-world measurements refers to developing hybrid models that combine physics-based simulations with machine learning-based corrections.

Close to both given-above threads is the predictive modeling of air jet behavior based on various parameters such as the pressure signals, intensity fields, and geometry features of the organ pipe. Hence, one can imagine that such a tool may first help to build a model to predict complex air flow patterns and phenomena not captured by traditional simulation methods and then build a pipe organ based on such a model.

We also investigated the use of machine learning for the real-time monitoring and control of air flow in organ pipes. This can involve developing algorithms that analyze sensor data in real-time to detect and mitigate issues such as turbulence or irregularities in air flow.

There are also other applications that are more closely related to sensor-based measurements that could be considered regarding experimental design optimization. Developing machine learning techniques to integrate data from multiple sensors, such as acoustic vector sensors, intensity probes, and high-speed cameras, can provide a more comprehensive understanding of air flow phenomena. This could involve fusion algorithms that combine data from different sources to improve overall accuracy and reliability. Moreover, using predictive models may help in identifying optimal sensor placement, measurement configurations, and experimental conditions to maximize information gain, reduce artifacts, and minimize resource utilization. Finally, the sensor-based approach needs the quantification of uncertainty and sensitivity analysis. Utilizing machine learning techniques for uncertainty quantification and sensitivity analysis to assess the impact of various factors (e.g., geometry, boundary conditions) on air flow predictions may help to identify critical parameters and sources of uncertainty in the modeling process. To facilitate solving some of the above problems, machine learning can be applied. 

## Figures and Tables

**Figure 1 sensors-24-01962-f001:**
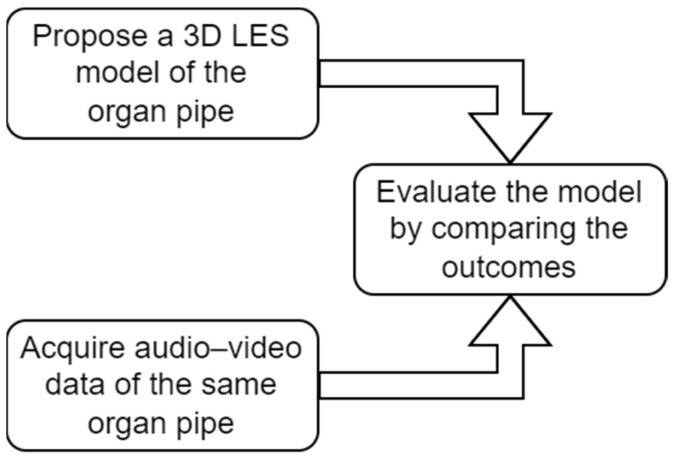
An outline of the study goals.

**Figure 4 sensors-24-01962-f004:**
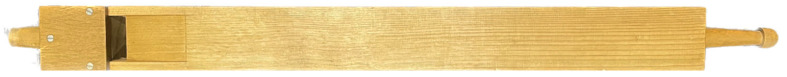
Bourdon pipe.

**Figure 5 sensors-24-01962-f005:**
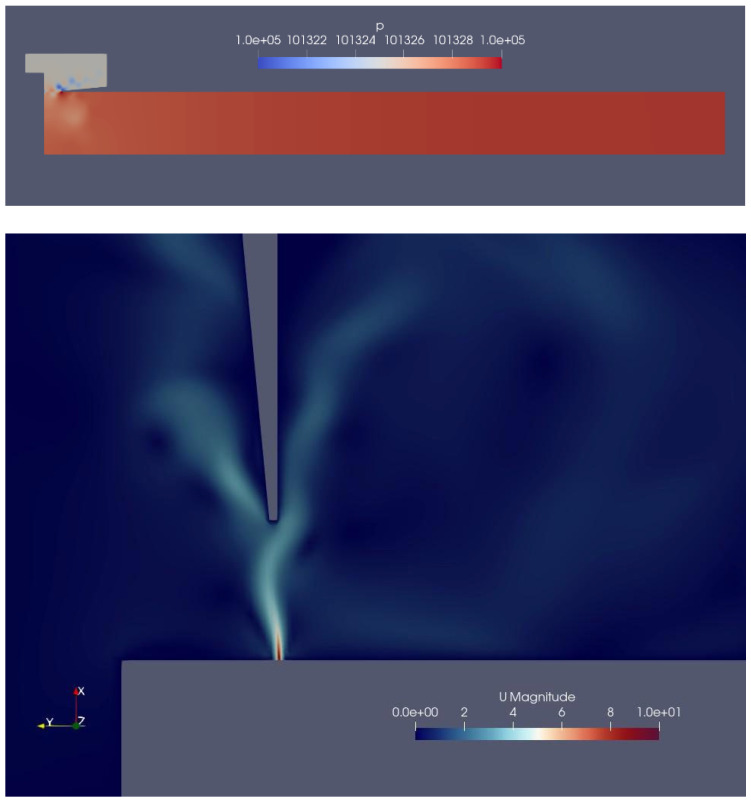
Pressure field (**top**) and detail of the velocity field at the mouth region (**bottom**). In both figures, the grey part represents the obstacles, i.e., the pipe walls and the far-field space outside the mouth. In the top figure (pressure field), the pipe lies flat and is looked at from its side; the red color of the resonator suggests that it was taken at the moment when the pressure reaches its maximum, and therefore, the jet is pointing inside the resonator, but it is being pushed outside it. In the bottom figure (velocity field), the pipe is standing, and a detail of the mouth is given; here, the blue background indicates that the air is not moving, except for a little sheet just outside the flue—the jet—and some turbulence is caused by the scattering of the jet into the environmental air.

**Figure 6 sensors-24-01962-f006:**
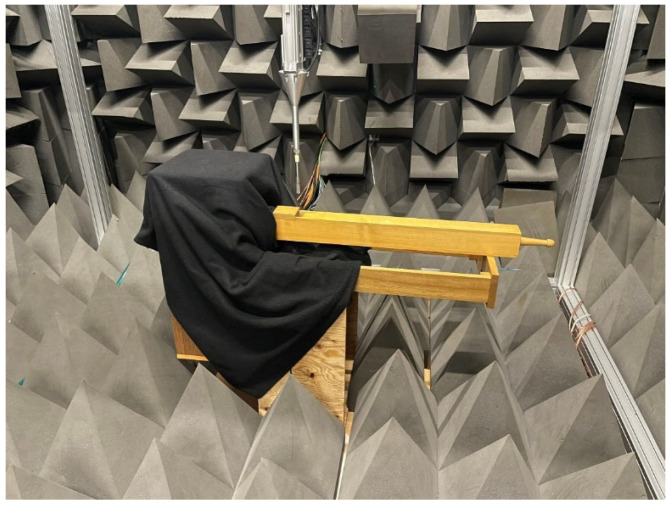
Intensity field measurement setup in the anechoic chamber. The pipe is placed horizontally to facilitate the Cartesian robot’s movements, and the base of the pipe, essential for the air supply, is covered with a sound absorber cloth to minimize the sound reflection caused by it.

**Figure 7 sensors-24-01962-f007:**
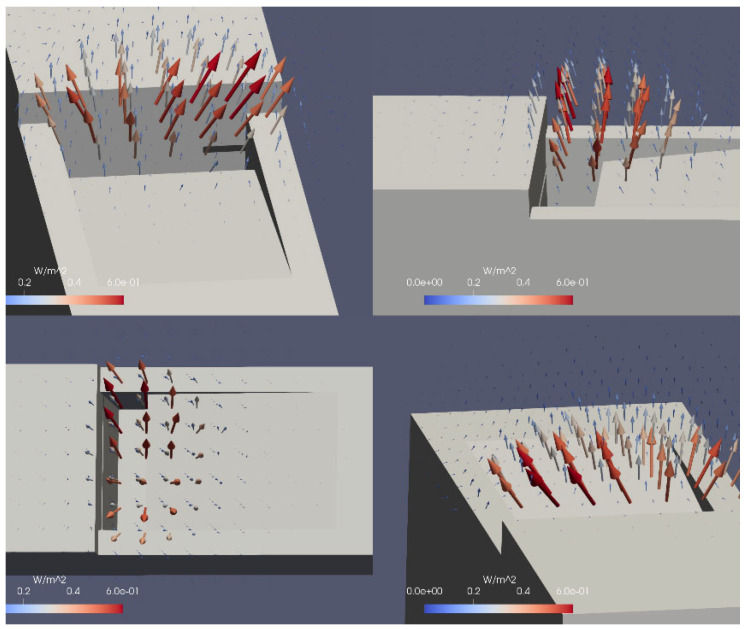
Different perspectives of the intensity field visualization. The color and size of the arrows depend on the magnitude of the sound intensity, and their direction indicates the precise position of the sound source, i.e., the edge of the labium.

**Figure 9 sensors-24-01962-f009:**
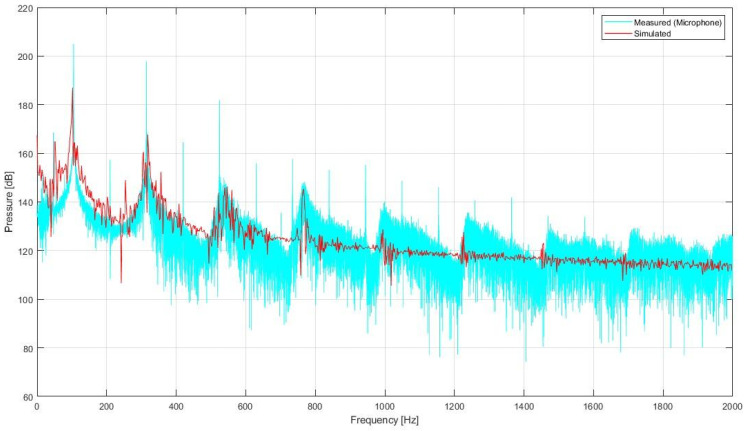
Comparison between the simulated pressure signal spectrum and the measured one. The latter clearly shows the superimposition of a series of overtones (the sharp peaks) emerging from the noise-like background modulated by the resonator and acoustic modes. Those modes are adequately reproduced by the signal coming from the simulation.

**Figure 10 sensors-24-01962-f010:**
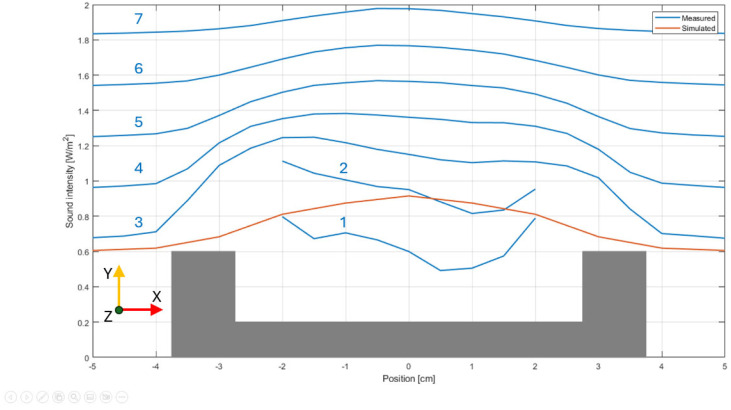
Measured (blue line) and simulated (red line) intensity distribution around the pipe’s mouth (indicated as grey part in the figure). The measured lines are shifted to increase the readability and resemble the position in the space of the probe’s path. From the bottom to the top, they were taken at the following distances from the labium plane: (1) 5 mm, (2) 10 mm, (3) 15 mm, (4) 20 mm, (5) 25 mm, (6) 30 mm, and (7) 35 mm. The simulated distribution has the same shift as line no. 3 (+0.6 W/m^2^).

**Figure 11 sensors-24-01962-f011:**
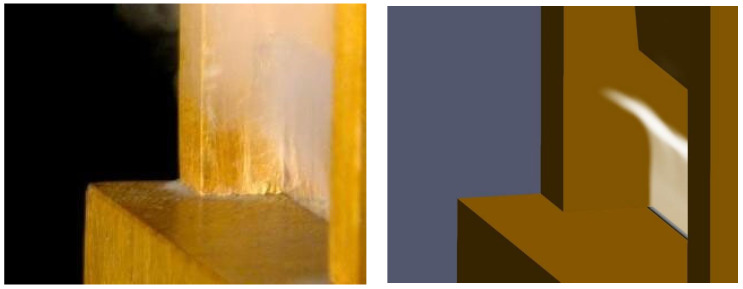
Still images of the jet displacement from the side perspective: HSC frame (**left side**); numerical simulation (**right side**). (See the YouTube link: https://youtu.be/WlrlTKOpdFo (accessed on 9 March 2024) to observe how the air jest is built).

**Figure 12 sensors-24-01962-f012:**
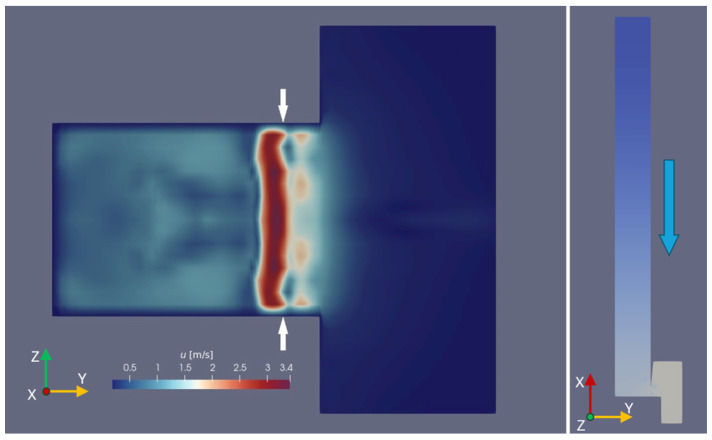
Jet shape seen from the labium perspective (i.e., from above, when the pipe is standing vertically). The velocity field shown refers to a horizontal plane lying at half the height of the pipe’s mouth. The white arrows indicate the vertical projection of the windway position; therefore, the left side of the picture represents a section of the air volume inside the resonator, while the right side is a section of the portion of free air involved in the simulation. This specific still image refers to the moment when the air jet points towards the pipe’s resonator, revealing the non-flat cross-section shape of the air jet.

**Table 1 sensors-24-01962-t001:** Boundary and initial conditions.

Boundary Type	Velocity	Pressure	Temperature
inlet	linearRamp	zeroGradient	fixedValue
outlet	inletOutlet	waveTransmissive	inletOutlet
wall	noSlip	zeroGradient	fixedValue
symmetry	symmetry	symmetry	symmetry
initial value	(0, 0, 0) m/s	101,325 Pa	293 K

## Data Availability

Data are contained within the article.

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
