# Peer review of "Sounding Mechanism of a Flue Organ Pipe—A Multi-Sensor Measurement Approach"

_sensors, 2024, doi:10.3390/s24061962_

Round 1
Reviewer 1 Report
Comments and Suggestions for Authors
Dear Authors,
the manuscript is well organised and presents a set of measurements carried out with a pressure-particle velocity probe. These measurements have been compared with numerical simulations.
The topic is of interest for the part of the scientific community interested in the acoustics of musical instruments. Still, the manuscript must be improved before it can be considered for publication.
- Abstract: What is a "cartesian robot"? Better specifying the type of robot used and how it works in the following sections.
- Introduction: is the flue organ pipe emission mechanism really related to vibro-acoustic problems?
Lines 31, 47, 48, 125, 126, 129, 146, 173: please avoid group citations.
Line 38: Please add citations to the papers considering non linear interactions solved by numerical models.
Lines 69-78: Why use roman numbers for the sections when they are numbered with arabic numbers?
- Section 2: line 81, is the term "rank stops" the correct one? Can you add them in Figure 1?
- Section 3: line 200, is "obstacle" a type of boundary condition? Please, change the term.
You completely left out the details about the model grid and other inputs. This section must be expanded
- Section 4: It is quite evident that the probe used is a Microflown. To my knowledge, this type of probe has some problems in the evaluation of the particle velocity when exposed to an air flow. How did you solve this issue, especially making some measurements so close to the outlet?
- Section 5: "When compared with the simulated signal, it can 279 be observed that the simulated fundamental frequency (102 Hz) shows good agreement with the real one (104.9 Hz)." Where (in what position) was the measurement performed?
Line 294: What about the amplitude of the simulated spectrum? Whas it corrected?
- Section 6: Lines 405-407, machine learning was never mentioned before. Please, expand the idea or leave it out.
The manuscript needs moderate editing in the English language.
Author Response
First, we would like to thank the Reviewers for their valuable suggestions and analysis of our paper. In the file, we present each reviewer's comment, followed by our response (in blue).
Also, changes required to the text are in blue font in the revised text of our paper. In the Answer to the reviewer file, we have copied some text from the revised paper that directly answers the Reviewer's comments.

Reviewer 2 Report
Comments and Suggestions for Authors
The authors presented a comprehensive flow and acoustics measurement of a flue organ pipe. Although investigation of the sound generation mechanism of a flue organ pipe is not a new topic in the acoustic community, employing various sensor techniques allows one to gain further insights into the underlying physics of a real musical instrument. To this end, the work contains a certain amount of novelty that will advance current knowledge.
However, it is this reviewer's opinion that a considerable improvement should be made to the manuscript. A large portion of the manuscript was spent on describing the setup of the experiment. Although this is necessary for understanding the details of the experiment, it, in fact, makes no scientific contribution. The reviewer expects to see an in-depth analysis of measurement data and gain further insights into the sound generation mechanism in Section 5. However, the discussion presented was too superficial. The authors should make a significant improvement of this section before the manuscript is recommended for publication.
Comments on the Quality of English LanguageLine 174, large-scale fluctuations, what fluctuation does it refer to? pressure?
Line 277, with respect to the pressure signal, you should specify the position where the signal is obtained.
Author Response
To the Reviewer,
First, we would like to thank the Reviewers for their valuable suggestions and analysis of our paper. In the file, we present each reviewer's comment, followed by our response (in blue).
Also, changes required to the text are in blue font in the revised text of our paper. In the Answer to the reviewer file, we have copied some text from the revised paper that directly answers the Reviewer's comments.

Reviewer 3 Report
Comments and Suggestions for Authors The authors present the study of experimental and simulation measurements of air flow and sound on an organ pipe. The paper is well written, the results are novel, and the presentation is mostly clear. I think however the authors need to put more effort in the presentation of the results: they clearly understand what they are seeing but don't manage to translate it in a clear way so others can understand it. The captions of the figures are in particular quite poor, just one sentence that works more as a title. Captions should be detailed enough to understand the figure without recourse to the main text, where they should be explained and interpreted. This is missing from the current version of the article. Adding video material would be also excellent as their description of the video is not particularly clear. In general, an excellent work and I'm looking forward to the revised version of the article. Detailed comments:L75 "findings concluded", it sounds weird in english.
L98 could the authors provide a more modern reference to this claim? I personally find there is a lot of repeated knowledge that has not been necessarily being replicated with modern research methods. If not in pipe research at least for other musical instruments. L111 20th century, not XX L112 ditto L125 hyphenation of analysis L158 flue -> flute L173 seems there's an extra space before "in" L176 the word energy has a smaller font L177 "stopped" small font L196 how does the mesh size compare to the oscillations of the flow? Fig. 8 Would not be easier to distinguish if the data was smoothed a bit? It's not clear which mode is wich, adding a mark to the plots where the modes are would make the things more clear. It is also not clear what are the modes and what are the partials of the sound. It's not clear though why the simulation should not reproduce the latter. L292 what are those "some undesired resonances"? Fig 9 The analysis of this figure is rather confusing. The lines are not contourns in teh traditional term, one uses contours to display iso-lines, which is not the case here. They analyse first the experimental data, and then the simulation, but this is not clear in the text. Why is there only one simulation results line? What are the different measurment lines? different Z coordinates? this is never explained. Please re-do this figure and use colors to show what's varying in the different experimental lines, as it stands the pressure is polyvalued, for each position there are several pressures at the same point which is unphysical. Fig10 I cannot see what the authors are describing, I cannot see anything meaningful in the left image of fig. 10. Maybe some lines and diagrams would help? Otherwise I would remove this figure from the paper as I don't think it adds anything meaningful, just anecdotal data difficult to precisely describe. Fig. 11 the authors needs to explain better this figure, what's the plane we are looking at? The axes how do they relate to the previous figure (which doesn't have axes btw)? at what location of the flute? Can the authors show the video files? Comments on the Quality of English Languagesee detailed comments, just a few typos
Author Response
To the Reviewer,
First, we would like to thank the Reviewers for their valuable suggestions and analysis of our paper. In the file, we present each comment made by the Reviewer, followed by our response (in blue).
Also, changes required to the text are in blue font in the revised text of our paper. In the Answer to the reviewer file, we have copied some text from the revised paper that directly answers the Reviewer's comments.

Round 2
Reviewer 1 Report
Comments and Suggestions for Authors
Dear Authors,
My remarks have been correctly captured by the improvements you made.
Best regards
Reviewer 2 Report
Comments and Suggestions for Authors
The manuscript can be accepted in the current form.